# OpenReview forum: "Causal Shapley Values via Bayesian Structure Learning: Provably Faithful Explanations for Black-Box Models"
_mathai.club/MathAI/2026/Conference — Submitted to 2026_

### Official Review · Reviewer_DiTj · 2026-03-10
**Strong Reject: Important Topic but the Current Manuscript Contains Fundamental Issues in Theory, Method Definition, and Experimental Credibility**

**Rating:** 2
**Confidence:** 4

**Review:**

# Title

**Strong Reject: Important Topic but the Current Manuscript Contains Fundamental Issues in Theory, Method Definition, and Experimental Credibility**

# Review

## 1. Summary

This paper proposes **CAUSALSHAP**, aiming to integrate Bayesian causal structure learning with Shapley-value-based explanations. The core idea is to replace the empirical conditional distributions used in standard SHAP with intervention distributions derived from a learned causal graph. The authors claim theoretical guarantees including faithfulness, efficiency, and consistency, and report empirical improvements over KernelSHAP, TreeSHAP, and Marginal Shapley on synthetic data, MIMIC-III, and German Credit datasets.

The motivation is meaningful: when features have causal dependencies, standard conditional SHAP explanations may indeed produce misleading attributions. The attempt to connect feature attribution with the underlying data-generating mechanism is conceptually interesting and potentially impactful.

However, despite the interesting motivation, the current manuscript contains several **serious issues in methodology, theory, and experimental validation**, which significantly undermine the credibility of the claims.

---

# 2. Strengths

1. **Important problem setting.**
   The distinction between correlation-based explanations and causal explanations is highly relevant, especially in high-stakes domains such as healthcare and finance.

2. **Reasonable conceptual motivation.**
   Moving from the standard SHAP value function

$$
v_{\text{SHAP}}(S) = \mathbb{E}[f(x)\mid x_S = x_S^*]
$$

to a causal-intervention-based value function is a sensible direction.

3. **Complete high-level structure.**
   The paper includes sections on method, theory, algorithm, experiments, and related work, which makes the intended narrative relatively clear.

---

# 3. Major Weaknesses

## (A) Methodological Definition is Inconsistent and Not Fully Rigorous

There is an inconsistency in how the prior over graphs is described. The abstract and introduction claim the use of a **Dirichlet–Laplace DAG prior**, whereas Equation (9) in the methodology corresponds to **independent Bernoulli edge priors combined with Laplace priors on coefficients**. These represent different model assumptions.

Furthermore, the variational approximation

$$
q(G) = \prod_{i \neq j} q_{ij}(e_{ij})
$$

assumes independent edges. Such a mean-field approximation **does not guarantee acyclicity**. Without explicit constraints enforcing DAG structure (e.g., NOTEARS-style constraints, ordering-based parameterization, or projection methods), samples from this distribution will generally produce arbitrary directed graphs rather than DAGs.

However, Algorithm 1 states that a DAG is sampled from $q(G)$ using the Gumbel-max trick without explaining how acyclicity is enforced. This omission is critical, since the validity of the method depends on working with DAGs.

However, Equations (6), (11), (12), and Algorithm 1 do not form a coherent estimator. In particular:

* Equation (12) replaces a distribution-level expectation with a plug-in estimate $f(\hat{x}_{\text{post-int}})$.
* The algorithm computes

$$
f(x_{\text{post-int}}) - f(x_{\text{post-int}}, -i)
$$

but the operation $f(x_{\text{post-int}}, -i)$ is not defined.

Overall, the manuscript does not clearly distinguish between:

* the causal value function
* the intervention distribution
* the Monte Carlo estimator
* the final Shapley attribution

This lack of formal clarity is problematic for an explanation-method paper.

---

## (B) Theoretical Claims Are Currently Unreliable

The theoretical section contains serious issues.

Theorem 1 introduces

$$
d_{\mathrm{TV}}(\hat{\phi}_i^{\text{causal}}, \phi_i^*)
$$

as a total variation distance. However, $\phi_i$ is defined as a **scalar attribution value**, not a probability distribution. Therefore, the use of total variation distance is mathematically inappropriate unless a distribution over attributions is explicitly defined.

The theorem further claims

$$
\mathbb{E}!\left[d_{\mathrm{TV}}(\hat{\phi}_i^{\text{causal}}, \phi_i^*)\right]=O!\left(n^{-1/3} + \frac{1}{\sqrt{NM}}\right)
$$

under unspecified “regularity conditions”. However, the manuscript does not provide sufficient assumptions to justify such a convergence rate.

For posterior concentration in causal structure learning, one would typically require conditions such as:

* identifiability assumptions
* faithfulness or minimality conditions
* prior support on the true DAG
* sparsity assumptions
* dimensionality scaling conditions

None of these are formally specified, and the proof sketch appears incomplete.

Similarly, the complexity analysis is inconsistent with the algorithm. The paper states the complexity as

$$
O!\left(2^p \cdot \text{poly}(n,p)\right)
$$

yet the algorithm actually relies on stochastic sampling of DAGs and permutations. Therefore, the runtime should depend on the number of Monte Carlo samples rather than on full subset enumeration.

---

## (C) Experimental Section Contains Major Inconsistencies

Several figures and captions appear inconsistent with the content shown.

Examples include:

* **Figure 3**: The caption describes posterior probabilities of true causal edges versus sample size, while the axes show permutation samples versus explanation MAE.
* **Figure 4**: The caption claims to demonstrate an $O(n^{-1/3})$ convergence rate, but the horizontal axis lists structural motifs rather than sample size.
* **Figure 2**: The caption claims the figure shows the learned causal structure for German Credit, but the diagram contains generic motifs such as Linear Chain, Fork, Collider, and Diamond.

These inconsistencies suggest that figures and captions were not carefully verified.

Another major issue arises in **Table 2**, which reports *Causal Alignment (Kendall–$\tau$)* with “true causal effects” for the MIMIC-III dataset. However, true causal effects are generally unavailable in observational medical datasets. The paper does not explain how these ground-truth values were obtained.

Finally, the claimed robustness to unobserved confounders is only described conceptually. The paper does not specify:

* the sensitivity model
* the confounding parameters
* identification bounds
* the experimental protocol

---

## (D) Literature Positioning is Weak

The related work section understates existing research on causal explanations and interventional attribution methods. The manuscript suggests that the correlation-versus-causation issue in SHAP explanations is “largely unrecognized”, which is not accurate.

The paper should more clearly clarify what constitutes its primary novelty:

* a new estimand
* a new causal structure learning framework
* a new statistical guarantee

At present, the contribution appears to be a combination of structure learning and intervention-based SHAP evaluation without clearly articulated novelty.

---

# 4. Quality, Clarity, Originality, Significance

**Quality**

The overall quality is insufficient for acceptance. Both theoretical claims and empirical results require substantial revision.

**Clarity**

While the high-level narrative is understandable, the technical exposition lacks precision. Notation reuse, undefined operators, and inconsistent equation references create confusion.

**Originality**

The research direction is promising, but the manuscript does not convincingly demonstrate a clear novelty relative to existing causal attribution methods.

**Significance**

If a rigorous framework combining causal structure learning and Shapley explanations were developed, it could be impactful. However, the current manuscript does not yet achieve this.

---

# 5. Pros and Cons

## Pros

* Important research problem
* Conceptually interesting idea of causal Shapley attribution
* Clear high-level motivation

## Cons

* Variational posterior does not guarantee DAG structure
* Method definition is incomplete and inconsistent
* Theoretical results are not mathematically well-defined
* Complexity analysis does not match the algorithm
* Experimental figures contain serious inconsistencies
* Real-world experiments lack credible causal ground truth
* Confounding robustness claims are unsupported
* Weak literature positioning

---

# 6. Final Recommendation

In its current form, the paper should **not be accepted**. While the problem is interesting and the direction promising, the manuscript currently resembles a research proposal rather than a fully validated research contribution.

---

# Rating

**2: Strong rejection**

# Confidence

**4: The reviewer is confident but not absolutely certain that the evaluation is correct**

---

### Official Review · Reviewer_PSD8 · 2026-03-12
**A weak point is the gap between theory and practice.**

**Rating:** 3
**Confidence:** 4

**Review:**

Authors propose a novel method for explaining machine learning model predictions — CausalSHAP, which combines Bayesian structure learning of causal relationships with the computation of Shapley values. Unlike classical SHAP methods, which assume feature independence, CausalSHAP takes into account the real causal dependencies between features, allowing for more reliable and interpretable explanations.
The main methodological problem of this method is that the work relies on a standard but very strong set of assumptions for causal identification. First, the authors explicitly assume that there are no unobserved variables simultaneously affecting two or more variables in the system. In real-world tasks (medicine, economics), this assumption is almost always false. Second, since the real world is full of feedback loops, the requirement of a directed acyclic graph (DAG) severely limits the applicability of the model. And third, it is assumed that the data is indeed generated by a linear Gaussian model. If the real process is non-linear or the noise is non-Gaussian, the graph estimates and, consequently, the Shapley values will be biased.
A strong point of the work is the elegant mathematical idea (replacing conditioning with do-calculus) and three rigorous theorems.
A weak point is the gap between theory and practice. In practice, the method requires overly ideal conditions (linearity, Gaussianity, absence of hidden variables) to be a reliable tool for explaining real complex models. Without solving the problem of hidden variables and scaling, the method risks remaining an academic exercise.

---

### Official Review · Reviewer_eT3o · 2026-03-12
**An attempt of faithful, efficient, and consistent framework with unverifiable results**

**Rating:** 3
**Confidence:** 4

**Review:**

The authors claim a new framework and three theorems.

$Claimed \ contributions \ (reformulated)$

0. CausalSNAP framework
1. Faithfulness Theorem depicting convergence to true causal effects.
2. Efficiency Theorem as of $O(2^p \cdot poly(n)$ time, for number of features $p$.
3. Consistency Theorem guarantees convergence to ground-truth.

$ Originality/Novelty$

The results appear novel. The subject is an interesting attempt at a causal generalization of the SNAP framework.

$Soundness/Significance$

While context and definitions are presented nicely, the authors provide only sketches of the proofs for all theorems (the reviewer expected them in the Appendix, but none were found). Proof sketches? These are typically used for illustration or to develop a reader's intuition.

For Theorem 1, that approach is clearly insufficient.

Theorem 2 gives a good idea of a proof.

For Theorem 3, it is not sufficient at all. Immediately, the statement in line 261 does not follow directly from Schwartz's theorem, as it requires a few additional assumptions.

As a result, the reviewer is left with incomplete material that is potentially promising but, in its current form, unverifiable.
For that reason, one cannot talk about significance.

$Weaknessess$

Absence of complete proofs results in diminished significance and soundness.

$Strengths$

The subject is important for causal ML.

$Conclusion$
Despite the obvious promise, the results are unverifiable in their current form. Hence, the conclusion.

---

### Decision · Program_Chairs · 2026-03-14

**Decision:**

Reject

**Comment:**

After careful evaluation by the Program Committee, we regret to inform you that your submission has not been accepted for presentation at MathAI 2026.

All submissions underwent a rigorous two-stage review process. Unfortunately, the reviewers identified one or more of the following concerns with your paper:

- Insufficient mathematical rigor or novelty relative to the existing body of work in the field;
- Presentation of results that substantially overlap with or rephrase previously published findings without clear original contribution;
- Significant issues with technical quality, including but not limited to broken or non-existent references, unsupported claims, or methodological gaps;
- Indications that the manuscript may have been generated with the assistance of large language models without substantial original intellectual contribution by the authors.

We received a large number of submissions this year, and the selection process was highly competitive. We encourage you to carefully consider the reviewers’ feedback (available through OpenReview), revise your work accordingly, and consider submitting an improved version to a future edition of MathAI or to another appropriate venue.

We appreciate your interest in MathAI and hope you will continue to engage with the conference community.

With kind regards,

MathAI 2026 Program Committee
URL: https://mathai.club
Telegram: https://t.me/MathAI_club
Email: mathai.club@yandex.ru